# HCV Diagnosis and Sequencing Using Dried Blood Spots from Patients in Kinshasa (DRC): A Tool to Achieve WHO 2030 Targets

**DOI:** 10.3390/diagnostics11030522

**Published:** 2021-03-15

**Authors:** Teresa Carrasco, David Barquín, Adolphe Ndarabu, Mirian Fernández-Alonso, Marina Rubio-Garrido, Silvia Carlos, Benit Makonda, África Holguín, Gabriel Reina

**Affiliations:** 1Microbiology Department, Clínica Universidad de Navarra, 31008 Pamplona, Spain; mcarrasco.4@alumni.unav.es (T.C.); dbarquin@alumni.unav.es (D.B.); mferalon@unav.es (M.F.-A.); 2Department of Internal Medicine, Centre Hospitalier Monkole, 4484 Kinshasa, Democratic Republic of the Congo; ado.ndarabu@gmail.com (A.N.); benimakondabm@gmail.com (B.M.); 3ISTUN, Institute of Tropical Health, Universidad de Navarra, 31008 Pamplona, Spain; scarlos@unav.es; 4IdiSNA, Navarra Institute for Health Research, 31008 Pamplona, Spain; 5HIV-1 Molecular Epidemiology Laboratory, Microbiology and Parasitology Department and Instituto Ramón y Cajal para la Investigación Sanitaria (IRYCIS), Hospital Universitario Ramón y Cajal, CIBER en Epidemiología y Salud Pública (CIBERESP), Red en Investigación Translacional en Infecciones Pediátricas (RITIP), 28034 Madrid, Spain; marina0705@hotmail.com (M.R.-G.); africa.holguin@salud.madrid.org (Á.H.); 6Department Preventive Medicine and Public Health, Universidad de Navarra, 31008 Pamplona, Spain

**Keywords:** hepatitis C virus, dried blood spots, direct-acting antivirals, resistance-associated substitutions, Kinshasa, genotype 4

## Abstract

The World Health Organization has established an elimination plan for hepatitis C virus (HCV) by 2030. In Sub-Saharan Africa (SSA) access to diagnostic tools is limited, and a number of genotype 4 subtypes have been shown to be resistant to some direct-acting antivirals (DAAs). This study aims to analyze diagnostic assays for HCV based on dried blood spots (DBS) specimens collected in Kinshasa and to characterize genetic diversity of the virus within a group of mainly HIV positive patients. HCV antibody detection was performed on 107 DBS samples with Vidas^®^ anti-HCV and Elecsys anti-HCV II, and on 31 samples with INNO-LIA HCV. Twenty-six samples were subjected to molecular detection. NS3, NS5A, and NS5B regions from 11 HCV viremic patients were sequenced. HCV seroprevalence was 12.2% (72% with detectable HCV RNA). Both Elecsys Anti-HCV and INNO-LIA HCV were highly sensitive and specific, whereas Vidas^®^ anti-HCV lacked full sensitivity and specificity when DBS sample was used. NS5B/NS5A/NS3 sequencing revealed exclusively GT4 isolates (50% subtype 4r, 30% 4c and 20% 4k). All 4r strains harbored NS5A resistance-associated substitutions (RAS) at positions 28, 30, and 31, but no NS3 RAS was detected. Elecsys Anti-HCV and INNO-LIA HCV are reliable methods to detect HCV antibodies using DBS. HCV subtype 4r was the most prevalent among our patients. RASs found in subtype 4r in NS5A region confer unknown susceptibility to DAA.

## 1. Introduction

Viral hepatitis is a global public health problem. According to the World Health Organization (WHO), in 2015 there were 71 million people worldwide with chronic hepatitis C virus (HCV) infection, corresponding to a global prevalence of 1.0%. Moreover, in the same year, there were 1.75 million new HCV infections reported. The proportion of diagnosed people all over the world is 20%, with an estimated 6% of patients diagnosed in low and middle-income countries (LMIC). Among those diagnosed, only 7% have started treatment (3.5% with direct-acting antivirals (DAAs) [1].

While the HCV incidence is decreasing globally, the mortality is increasing, and there are 1.3 million deaths due to viral hepatitis each year. Of these figures, HCV complications such as cirrhosis and hepatocellular carcinoma were responsible for approximately 30% of deaths, accounting for a total of 475,000 due to HCV [1,2]. The United Nations and WHO have proposed a strategy to eliminate viral hepatitis by 2030. The elimination targets are to reduce 65% mortality and 90% global incidence of both hepatitis B and C from the 2015 baseline data [1]. To achieve these goals, WHO has defined some interventions that include diagnosing 90% of HCV individuals, treating 80% of them with DAAs, and reducing transmission by improving health-care procedures and harm reduction [2].

Sub-Saharan Africa (SSA) has a high burden of viral hepatitis, and there is a lack of quality data about prevalence and distribution of genotypes (GTs) in many countries [3]. It is estimated that globally 15% of people living with HCV (PLWHC) are in this region [4], accounting for a local HCV prevalence of 2.3% in SSA [4], and only 6% of infections have been diagnosed [1]. Genotypes are heterogeneously distributed throughout the world: GT 1 is the most prevalent (44%) and is found mainly in high-income countries, GT 3 represents 25% of all infections, and GT 4 (15%) is found predominantly in low-income countries [5].

HCV isolates circulating in SSA belong mainly to GTs 1 and 4. In particular, non-1a/b or non-4a/d subtypes are uncommon in high-income countries, but are responsible for 5.5 million infections in SSA (54% prevalence in the region) [6]. Treatment failure with DAA is associated with resistant associated substitutions (RAS) in the viral genome. These are amino acid substitutions in the NS3, NS5A, or NS5B regions that reduce drug activity, and may appear during treatment or exist naturally before treatment. The most frequent subtypes circulating in SSA are different from those found in other parts of the world, and their susceptibility to DAA is unknown [7]. However, the use of DAA for the treatment of HCV in SSA has been demonstrated to be successful against genotypes 1–5, demonstrating the feasibility of HCV elimination in resource-limited settings [8,9].

To meet the WHO 2030 elimination targets, the number of people diagnosed in SSA must be increased, circulating strains identified, and a greater proportion of the infected population treated. However, screening tests are not available or unaffordable for many countries and, in general, there is usually no suitable infrastructure to perform them. In this context, alternative models using rapid diagnostic tests (RDTs) or dried blood spots (DBS) are emerging as alternative tools to carry out diagnosis [10]. Genotyping in SSA is not widely available, so treatments cannot be scheduled by genotype when pan-genotypic treatments are not accessible. Furthermore, the genetic information of HCV in SSA is scarce, so the burden and distribution of subtypes and RAS is poorly understood.

The Democratic Republic of Congo (DRC) is the largest country in Central SSA. In this area, only Gabon and the Central African Republic have reported prevalence estimates [5] so there is uncertainty in the prevalence estimates for neighboring countries. According to the Polaris Observatory, there are 1.8 million infected people in DRC (prevalence 2.1%), although only 3% of patients have been diagnosed [11], and GT 4 represents 96.8% of infections [5]. Under these circumstances, the estimated year to reach the WHO elimination targets will be after 2050 [2].

Several studies have shown the presence of strains circulating in SSA that are less susceptible to DAAs. Depending on the subtype, the impact of RASs on susceptibility can vary. The treatment of subtype 4r infections are particularly challenging, as less than 60% of sustained virological response (SVR) 12 weeks after the end-of-therapy has been reported after DAA treatment [12,13]. Several studies have found that baseline RAS located at positions 28, 30, and 31 of subtype 4r NS5A may explain reduced susceptibility to NS5A inhibitors [12,14]. In addition, post-treatment failure to NS5B inhibitors has been described in subtype 4r due to S282C/T mutation in viral RNA polymerase [14].

In the context of trying to reach WHO elimination targets, and focusing on the Democratic Republic of Congo, the aim of this study was to evaluate the feasibility of Dried Blood Spots (DBS) to carry out HCV serological and molecular analysis, including HCV-RNA detection, genotyping, and assessment of baseline RAS.

## 2. Materials and Methods

### 2.1. Study Design and Participants

The OKAPI (Observational Kinshasa AIDS Prevention Initiative) project is a prospective cohort study designed to evaluate factors associated with changes in HIV knowledge and sexual behaviors after 6 and 12-months of follow-up. From April 2016 to April 2018, people aged 15–59 years attending HIV Voluntary Counseling and Testing (VCT) at two reference hospitals in the urban area of Kinshasa were invited to participate. Those with a previous HIV positive test as well as pregnant women were excluded. Details of the study have been published elsewhere [15]. In addition, a group of 62 HIV negative not attending VCT were also recruited. No patient had previously been diagnosed with HCV disease and had not received prior HCV treatment.

### 2.2. Sample Collection

Dried Blood Spots (DBS) were collected from 364 adult patients attending Monkole and Kalembelembre Hospitals in Kinshasa (DRC) as part of the OKAPI project, which initially studied HIV infection in the area. From this collection, 270 DBS samples were randomly selected to investigate HCV infection through different tests. For sample collection, two Whatman 903 Protein Saver Cards (Schleicher & Schuell, Dassel, Germany) were prepared by spotting 70 µL of whole blood, collected by venipuncture in EDTA-anticoagulant tubes, into each spot. They were dried separately on a drying-rack overnight at room temperature, sealed in a zip-lock plastic bag with desiccant bags, and stored at −20 °C until transported in dry ice to Clinica Universidad de Navarra (Pamplona, Spain), where they were stored at −80 °C until further use.

Before serological or viral load testing, each spot was eluted with one milliliter of Phosphate-Buffered Saline (PBS) at 37 °C for 60 min. Of the spots collected in card 1, one was used for serological tests, one for HCV viral load assay, and two for molecular genotyping, while the second card was stored for further analysis.

### 2.3. Serological Tests

HCV serological tests, from 270 patients, carried out in this study were third-generation enzyme immunoassays (EIAs): Vidas^®^ anti-HCV (bioMerieux) (*n* = 171), Elecsys anti-HCV II (Roche Diagnostics, Basel, Switzerland) (*n* = 206), and INNO-LIA HCV Score (Fujirebio Europe) (*n* = 31). In this study, eluted DBS (1 spot in 1 mL of PBS during 60 min at 37 °C) was used for the determination of anti-HCV antibodies (Ab) instead of serum or plasma, based on the procedure described previously [16]. Samples were screened simultaneously on 107 samples for HCV seropositivity with Vidas^®^ anti-HCV and Elecsys anti-HCV II. In addition, 31 samples were also tested with INNO-LIA. Each assay was interpreted following manufacturer instructions for serum/plasma. Sensitivity and specificity of each assay was assessed considering that those patients with at least two positive/indeterminate tests among the three serological techniques used were true positives.

#### 2.3.1. Vidas^®^ Anti-HCV

Vidas^®^ Anti-HCV is an enzyme-linked fluorescent assay (ELFA) for the detection of anti-HCV Ab. For this assay, 100 microliters of 171 DBS specimens were tested as required by the instrument. The results were calculated automatically by the instrument, considering positive values those with a test value (VT) ≥ 1 and negative if VT < 1.

#### 2.3.2. Elecsys Anti-HCV II

The Elecsys anti-HCV II is an electrochemiluminescence immunoassay (ECLIA) for the detection of HCV antibodies using HCV core, NS3, and NS4 antigens. Fifty microliters (instrument preset volume) of 206 DBS specimens were evaluated. The system automatically gives the result of each sample as reactive (cutoff index (COI) > 1.0), non-reactive (COI < 0.9), or indeterminate (COI 0.9–1.0).

#### 2.3.3. INNO-LIA HCV Score

INNO-LIA HCV Score is a Line Immune Assay that detects HCV Ab with antigens (E2, NS3, NS4A, NS4B, and NS5A) coated in strips. Eighty microliters of eluted DBS were added to each strip to perform the assay following the manufacturer procedure for the 16 h incubation protocol. Results were interpreted following manufacturer instructions.

### 2.4. HCV RNA Detection: COBAS^®^ AmpliPrep/COBAS^®^ TaqMan^®^ HCV Test

HCV RNA quantification was performed with COBAS^®^ AmpliPrep/COBAS^®^ TaqMan^®^ HCV Test (Roche Diagnostics GmbH). Twenty-six samples were tested using 1000 µL of eluted DBS (1 spot eluted in 1000 µL of PBS during 60 min at 37 °C). The adjusted HCV viral load in plasma was calculated as previously described [17,18], taking into account the dilution factor of the elution process, the usual hematocrit values (42% for women and 47% for men), and considering that 70 µL of blood was collected per spot. The lower limit of RNA quantification following this methodology was estimated at 860 IU/mL of eluted DBS.

### 2.5. Molecular HCV Characterization

NS5A, NS5B, and NS3 regions from 11 HCV viremic patients were sequenced for genotype determination and resistance mutations analysis through in-house amplification protocols followed by Sanger sequencing.

#### 2.5.1. Primers Design

NS5B region was amplified following a modified pangenotypic nested PCR protocol [19], previously validated [20], yielding a fragment of 372 bp, encompassing codons 227–336 of the RNA polymerase. Then, specific NS3 and NS5A degenerated primers were designed based on previous reports [21] and available genomes for the specific subtypes detected in our series (4c/4k/4r). The NS3 region was sequenced following a nested PCR of 658 bp (codons 1–181), and NS5A region was amplified through a single PCR of 472 bp (codons 1–144). Primers previously designed for NS5B region and new primers for NS5A, and NS3 are shown in Table 1.

#### 2.5.2. RNA Extraction and Amplification

For HCV characterization, RNA was extracted from two DBS spots using manual High-Pure Viral Nucleic Acid (Roche) kit and reverse transcribed using RevertAid H Minus First Strand cDNA Synthesis Kit (ThermoFisher, Carlsbad, CA, USA) following manufacturer instructions.

Viral genome was amplified using the above primers as follows: The single (NS5A) or first-round (NS3/NS5B) PCR reaction was performed with 5 μL of cDNA for each target gene in a final volume of 25 μL reaction mix containing 1.5 pmol of each outer primers, 12.5 μL KAPA2G Fast HotStart ReadyMix (2X) (Sigma-Aldrich, Wilmington, MA, USA), and 4.5 μL water (Sigma-Aldrich). First-round amplification was carried out at 95 °C for 5 min and 45 cycles of denaturation at 95 °C for 10 s, annealing at 56 °C for 10 s, extension at 72 °C for 15 s, and a final elongation step at 72 °C for 1 min. One microliter of the first PCR product (NS3 and NS5B targets) was subjected to nested PCR with 1.5 pmol of each inner sense and antisense primers, 12.5 μL KAPA2G Fast HotStart ReadyMix (2X) (Sigma-Aldrich, Wilmington, MA, USA), and 8.5 μL water (Sigma-Aldrich) under the following amplification conditions: Denaturation at 95 °C for 5 min and 35 cycles of denaturation at 95 °C for 10 s, annealing at 56 °C for 10 s, extension at 72 °C for 15 s, and a final elongation step at 72 °C for 1 min.

#### 2.5.3. Sequencing and Interpretation

PCR amplicons were purified using the GFX™ PCR DNA and Gel Band Purification Kit (GE Healthcare, Chicago, IL, USA) and sequenced by CIMA Lab Diagnostics (Navarra, Spain). Sequences were edited with FinchTV and Clustal Omega to obtain FASTA files. Genotype/Subtype and resistance analysis was carried out using Geno2Pheno [HCV] website [22] and, finally, resistance-associated substitutions (RAS) interpretation was done following the rules published by Sorbo et al. [23].

Accession Numbers. NS3, NS5A, and NS5B HCV sequences were submitted to GenBank (www.ncbi.nlm.nih.gov/genbank, accessed on 18 August 2020) with the following accession numbers: MT893899-MT8938924.

### 2.6. Ethical Aspects

The project was approved by the Human Subjects Review Committees at Monkole Hospital/University of Kinshasa (Kinshasa, DRC) and University of Navarra (Pamplona, Spain). Informed consent of enrolled participants was obtained. All methods were carried out in accordance with relevant guidelines and regulations.

## 3. Results

### 3.1. HCV Serological Assays on DBS

Vidas^®^ Anti-HCV was performed on 171 DBS specimens and 11.7% tested positive. Among the 206 samples tested with Elecsys anti-HCV II, 5.8% were positive. Vidas^®^ Anti-HCV and Elecsys anti-HCV II were conducted simultaneously on 107 samples (56.9% female, mean age 42.3 ± 13.9 years, 74.8% co-infected with HIV), and INNO-LIA HCV was carried out on 31 of these 107 samples, which resulted in positive or indeterminate on either VIDAS or Elecsys platforms. Within this group of 107 samples, Vidas^®^ Anti-HCV tested positive in 18 samples (16.8%), Elecsys anti-HCV II in 12 specimens (11.2%), and INNO-LIA-HCV in 10 samples. Finally, HCV seroprevalence of the cohort was calculated 12.2% as 13 true positive results were obtained (patients with at least two positive/indeterminate tests among the three methods evaluated). All the results are summarized in Figure 1.

Table 2 shows the evaluation carried out for HCV serological assays on DBS. The sensitivity of Vidas^®^ Anti-HCV (61.5%) was lower than that of Elecsys Anti-HCV II and INNO-LIA HCV (100%). Of 94 Elecsys Anti-HCV II seronegative patients, 10 were positive with Vidas^®^ Anti-HCV (specificity of 89.4%), that is, false positive results were more frequent in Vidas^®^ Anti-HCV assay.

### 3.2. HCV RNA Detection on DBS

Twenty-three patients had at least one positive result with Vidas^®^ Anti-HCV, Elecsys anti-HCV II, or INNO-LIA HCV, and underwent RT-PCR, together with some randomly selected negative samples (Figure 1). Viral load of these specimens was quantified with COBAS^®^ AmpliPrep/COBAS^®^ TaqMan^®^. Eleven out of 107 samples (10.3%) from 13 patients with confirmed HCV infection had detectable viral RNA, as shown in Figure 1; HCV viral load values ranged from 860 to 423,645 IU/mL. The sensitivity of this molecular tool for HCV diagnosis resulted 84.6% and the specificity 100%.

### 3.3. HCV Genotype and Resistance Analysis

The 11 seropositive samples with detectable HCV RNA were selected to perform sequence analysis of NS3, NS5A, and NS5B regions. Subtype was established on 10/11 sequences derived from NS5B testing (one sample with a viral load below 1000 IU/mL did not amplify). All of them were identified as GT4 subtypes: 50% of patients were infected with subtype 4r, 30% with 4c, and 20% with 4k. In addition, 8 isolates yielded NS3 (Table 3) and NS5A (Table 4) amplification products.

Among NS3 sequences, no relevant polymorphisms at positions associated with resistance to NS3 protease inhibitors were observed (Table 3). However, all 4r isolates harbored two mutations: R26K and A181S; L14F and T122N mutation were present in two out of three 4c isolates (Table 5). These substitutions had not been previously described.

Within NS5A region, the most common positions associated with DAAs resistance in subtype 4r are 28, 30, 31, and 93 [23]. In our group of patients, there were no substitutions observed at position 93. The NS5A_28–32_ motif LPRM was not present in any of our 4r sequences. Among 4 samples analyzed for 4r isolates, all of them harbored NS5A RAS L28V/F/I; one contained R30S RAS, and all of them contained M31L substitution (Table 4). Among 4c and 4k subtypes, there was only one baseline RAS (M31L) in the 4k isolate. The remaining polymorphisms detected in NS5A region are shown in Table 5.

The assessment of NS5B region showed that S282C/T RAS was not present in any isolate. Finally, the polymorphisms detected in these three HCV regions (NS3, NS5A, and NS5B) are summarized in Table 5.

## 4. Discussion

New screening and diagnosis tools are required to meet the WHO 2030 elimination targets for HCV. With this aim, specimens collected with DBS may be a good alternative to conventional serum/plasma samples for diagnosis in LMIC. Another alternative to improve access to HCV testing are rapid diagnostic tests (RDTs), immunoassays that detect antibodies or antigens and give a result in less than 30 min. The WHO Guidelines for Hepatitis B and C Testing recommend the use of quality-assured RDTs in settings where there are no laboratory facilities, and the use of DBS samples in people with poor venous access and places without nearby laboratories [24]. Nowadays there is a lack of validated commercial assays for DBS workflow. However, according to various studies, the detection of HCV antibodies and quantification of HCV RNA using DBS samples can be very reliable [17,25]. Several studies have focused on the consistency of DBS as an analyte for HCV diagnosis. A meta-analysis about diagnosis accuracy of serological assays using DBS reported a pooled estimated sensitivity of 98% and specificity of 99% [26]. Soulier et al. revealed that HCV RNA levels in DBS were lower than in serum and all patients were correctly genotyped using DBS [27]. This information supports the validity of our results obtained from DBS samples and allow us to better understand and interpret our results. Our study population showed a 12.2% prevalence of HCV antibodies and 10.3% of HCV RNA, high figures associated with a high-risk population (74.8% HIV-coinfected).

Among the serological tests evaluated in this study, a marked decrease in the performance of Vidas^®^ Anti-HCV assay was observed, while both Elecsys Anti-HCV and INNO-LIA HCV showed excellent sensitivity and specificity in the detection of HCV antibodies when using DBS from DRC. Vidas^®^ Anti-HCV test showed a sensitivity of 61.5% and specificity of 89.4% in this study. In particular, 10 out of the 18 positive results obtained by Vidas method using DBS corresponded to false positive results, this is similar to recent data from seven SSA countries (including DRC) which detected up to 80% of false positive results among HCV blood donors when plasma samples were assayed by rapid detection tests [28]. Poor performance of HCV serological tests using serum has also been reported in previous studies in Uganda for Ortho HCV version 3.0 ELISA Test [29], Rapidtest^®^, and ADVIA Centaur HCV assay (Siemens) [30]. Davis et al. [7], also found significant differences in specificity for INNO-LIA HCV and Elecsys Anti-HCV serological assays in serum samples from patients from Uganda and DRC.

Most HCV enzyme immunoassays (EIA) are assessed with people from developed countries, such as North America and Europe, where GT1 is the most frequent genotype. EIAs usually use HCV core and nonstructural proteins from GT1 as antigens, but we have found that GT4 is more prevalent than GT1 in Kinshasa. Genotype diversity may have an impact on the sensitivity of serological assays, as recently reported by Davis et al. [7]. On the contrary, Scheiblauer et al. concluded in their report that the sensitivity of the evaluated assays did not show significant genotype-dependent differences [31]. An alternative to INNO-LIA HCV (Geenius HCV Supplemental Assay, BIO-RAD), an immunochromatographic assay able to confirm the presence of antibodies against HCV within minutes, has recently been commercialized, as opposed to the 20 h procedure of the INNO-LIA test. Seven samples from our collection were correctly tested and classified by Geenius HCV (data not shown), suggesting good diagnostic accuracy of this new method for the detection of GT4 HCV antibodies from DBS samples. Future studies are needed about the validity of diagnostic tests to accurately determine whether or not a patient is infected with rare subtypes of HCV. Furthermore, tests using DBS samples should be validated in LMIC to increase the percentage of people diagnosed in these areas and to achieve WHO targets for HCV elimination by 2050 [2]. Choosing the appropriate diagnostic tool is essential to correctly detect HCV infection and implement preventive and therapeutic strategies in SSA.

A high rate of active HCV infection was detected within this group of Congolese patients (77%) mainly HIV co-infected, similar to 74–88% reported in blood donors of Ghana [32]. Xpert^®^ HCV Viral Load (Cepheid) has been recently developed, an automated all-in-one molecular method for HCV RNA quantification. Nine positive samples from our series were tested through Xpert^®^ HCV (data not shown) and correctly detected as expected. This new technique could be an alternative for HCV RNA quantification for samples collected on DBS in SSA, as recently reported for HIV viral load testing [18].

The NS5B region was chosen to characterize the genotype and subtype of our patients, since there are universal primers pairs for the amplification of this region of any HCV subtype [19]. In this study, subtypes were determined for 90.9% of samples and 4c/4k/4r subtypes were prevalent. With this information, genotype-specific primers were designed for NS3 and NS5A regions and were found to be effective in amplifying 4c/k/r subtypes, together with additional GT4 strains (data not shown). Direct sequencing of PCR products using genotype-specific primers was successful in this study as the majority of samples yielded interpretable sequences. There were three samples that could not be sequenced into NS3/NS5A, which had HCV RNA concentrations below the lower limit of quantification of COBAS^®^ AmpliPrep/COBAS^®^ TaqMan^®^. These results are consistent with other studies that reported problems in sequencing samples of patients with low-level viremia and produced a similar portion of NS5B sequencing subtyped samples (72.2%) [27].

Drug susceptibility was analyzed for DAAs approved for clinical use against HCV. This analysis was carried out studying the presence of resistance-associated substitutions previously described [23,33,34] among our sequences. As mentioned above, several studies have reported that susceptibility to DAA varies according to HCV genotype or subtype. GT 4, and more specifically subtype 4r, is associated with a high rate of treatment failure [13,14,35]. Fourati et al. identified two to three NS5A RASs (at least one in position 28 or 30) among patients who failed treatment, conferring reduced susceptibility to recommended first-line HCV treatment regimens [35].

Susceptibility to NS3 protease inhibitors (asunaprevir, glecaprevir, grazoprevir, paritaprevir, simeprevir, vaniprevir, and voxilaprevir) was analyzed by comparing the most frequent RASs reported in treatment failure in GT4 patients [23] with the amino acids found in our sequences. No natural polymorphisms associated with resistance were observed in the NS3 region in the subtypes detected in this study.

The susceptibility of NS5A inhibitors (daclatasvir, elbasvir, ledipasvir, ombitasvir, pibrentasvir, and velpatasvir) was analyzed. Multiple NS5A RASs described by Sorbo et al. [23] prior to treatment were found in our sequences that may confer resistance or poor response to different DAAs. L28V RAS, present in 4r subtype, is associated with high-level resistance to ombitasvir and possible resistance in vivo to daclatasvir and ledipasvir. The wild-type amino acid at position 30 in GT 4 is arginine (R), widely present in this series and naturally associated with drug resistance, specifically, it is likely to produce resistance to ledipasvir and possibly daclatasvir. A patient infected with subtype 4r harbored R30S RAS that confers resistance to daclatasvir and ledipasvir, as the mutation can cause >100-fold change resistance in GT4 replicons [23]. In this study, we also identified the M31L RAS that contributes to the resistance of 4r strains to daclatasvir, ledipasvir, and ombitasvir. A RAS pattern, previously described for ledipasvir, was observed in one of our sequences: L28V + L30R, which confers a 100-fold change resistance in GT 4 replicons. All isolates belonging to subtype 4c harbored 31M codon, which has been recently demonstrated to induce high-level resistance in subtype 4r for ledipasvir (2100-fold change in half maximal effective concentration) [36]. These authors manufactured replicons expressing the NS5A protein from unusual HCV subtypes (1l, 3b, 3g, 4r, 6u, 6v), which were resistant at some level to all NS5A inhibitors other than pibrentasvir [36]. Anyway, all our isolates might be susceptible to pangenotypic regimens based on pibrentasvir or velpatasvir, since these two drugs seem to be more effective against NS5A RAS at position 28, 30, and 31 [35], however, the availability of these pangenotypic regimens in SSA is still limited.

NS5B inhibitors include sofosbuvir and dasabuvir. The most common regimens are based on sofosbuvir, and its resistance is usually associated with S282T RAS. In this study of treatment naïve patients, no amino acid change was found at this position associated with reduced susceptibility. However, a previous study showed that S282T variant was detected in half of the patients infected with subtype 4r after treatment with sofosbuvir and who did not achieve SVR [14].

A number of NS3/NS5A mutations shown in Table 5 have not been previously described in GT 4, however, their effect on DAAs susceptibility is unknown. Further follow-up of DAA treated patients and characterization of substitutions associated with resistance may explain the effect of other polymorphisms reported in our study.

Our study shows, consistently with other authors, that subtype 4r strains may be resistant to regimens recommended for GT 4, especially to NS5A inhibitors because there might be RASs with major clinical impact [37]. Thus, HCV genotype and subtype should ideally be determined before treatment in regions where these HCV subtypes are present in substantial proportions, or in migrants from these regions, to optimize treatment regimens.

The selection of the therapy regimen for HCV is assessed by genotype determination, liver status, interactions with other drugs, and duration of the treatment. For instance, the European Association for the Study of the Liver (EASL) recommendations on treatment of hepatitis C [38] suggest different first-line therapies depending on the genotype, but not the subtype. There are currently four treatment options for patients infected with GT 4: Sofosbuvir/velpatasvir; glecaprevir/pibrentasvir; sofosbuvir/ledipasvir; and grazoprevir/elbasvir. Of these treatments, our patients are likely to fail sofosbuvir/ledipasvir. Re-treatment for patients who failed previous therapies is based on a triple DAA pangenotypic combination of sofosbuvir, velpatasvir, and voxilaprevir, not available at this moment in DRC. Our patients would be susceptible to this therapy according to the drug susceptibility analysis mentioned before. Many pangenotypic regimens include NS5A inhibitors, so further evidence of their effectiveness against the different HCV GT4 and non-1a/1b subtypes that infect patients in SSA is needed. Based on the experience in Rwanda, in the absence of advanced diagnostics to assess GT subtype, patients under 40, with a history of hospital admissions and surgeries, and advanced liver disease, may warrant closer follow-up for treatment failure or alternative DAA regimens since there is a high association with infections by subtype 4r [39].

In this study, HCV resistance tests were carried out using an in-house technique that sequences the isolates and checks for substitutions at positions associated with resistance previously described by Sorbo et al. [23]. There are no standardized methods to study HCV resistance. The EASL recommendations suggest testing for HCV resistance when SVR is not achieved to tailor second line therapy [38], while the American Association for the Study of Liver Diseases (AASLD) guidelines recommend RAS testing, if available, for subtypes 1a or 3 [40]. Recommendations in UK suggest searching NS5A RAS in subtypes not commonly found in high-income countries, including GT 4, 5, and 6 [41]. There is no RAS information regarding GT4 subtypes other than 4a/4d available in the American or European guidelines [40].

The importance of understanding the reduced susceptibility to DAA among HCV strains circulating in DRC is to tailor DAA regimens based on the genetic diversity of HCV in SSA, since, in general, clinical trials of DAA for HCV have not been carried out in African countries and infections caused by GT4 other than 4a/4d have been poorly evaluated [42]. In addition, validated information should be obtained to correctly treat infections caused by 4r/4c/4k subtypes affecting migrant patients in high-income countries. The high prevalence of GT4 in SSA warrants further studies to assess the sensitivity of diagnostic tests and validate effective HCV treatments in this geographic region, particularly when interferon-free DAA regimens become a reality.

Our study has several limitations. First, we did not evaluate diagnostic methods with plasma samples collected in DRC, as paired plasma/DBS specimens were not collected in the study population. Furthermore, the sample size in the evaluation of some tests, as INNO-LIA HCV, was low, so larger studies are needed to better determine the sensitivity and specificity of these commercial assays, especially in direct comparison with plasma/serum. Second, no next-generation sequencing (NGS) was performed, however, Sanger sequencing is accepted as a reliable tool for genotype and HCV RAS detection. Nevertheless, NGS sequencing for S282T detection among minority variants of HCV 4r strains could give some useful information. Third, storage conditions could influence the analysis, however, DBS specimens were transported on dry ice and preserved at −80 °C until use. Despite these limitations, DBS sampling has proven useful for diagnosis and characterization purposes. Furthermore, to our knowledge, there are no previous studies that have included a full description of HCV NS3, NS5A, and NS5B RAS, and polymorphisms in HCV strains circulating in DRC.

In conclusion, the WHO eradication plan will not be achieved by 2030 if people living in low- and middle-income countries cannot access diagnostic tests and subsequent treatment. DBS samples can be used to improve access to care, for both diagnosis and characterization. In particular, Elecsys Anti-HCV and INNO-LIA HCV seem reliable methods for detecting HCV antibodies using DBS collected in sub-Saharan Africa. In the present study, NS5B sequencing revealed exclusively GT4 isolates belonging to subtypes 4c, 4k, and 4r. HCV subtype 4r with unknown susceptibility to direct-acting antivirals due to frequent polymorphisms at positions 28, 30, and 31 of NS5A region was the most prevalent strain among HCV patients in Kinshasa, a city of 14 million inhabitants.

## Figures and Tables

**Figure 1 diagnostics-11-00522-f001:**
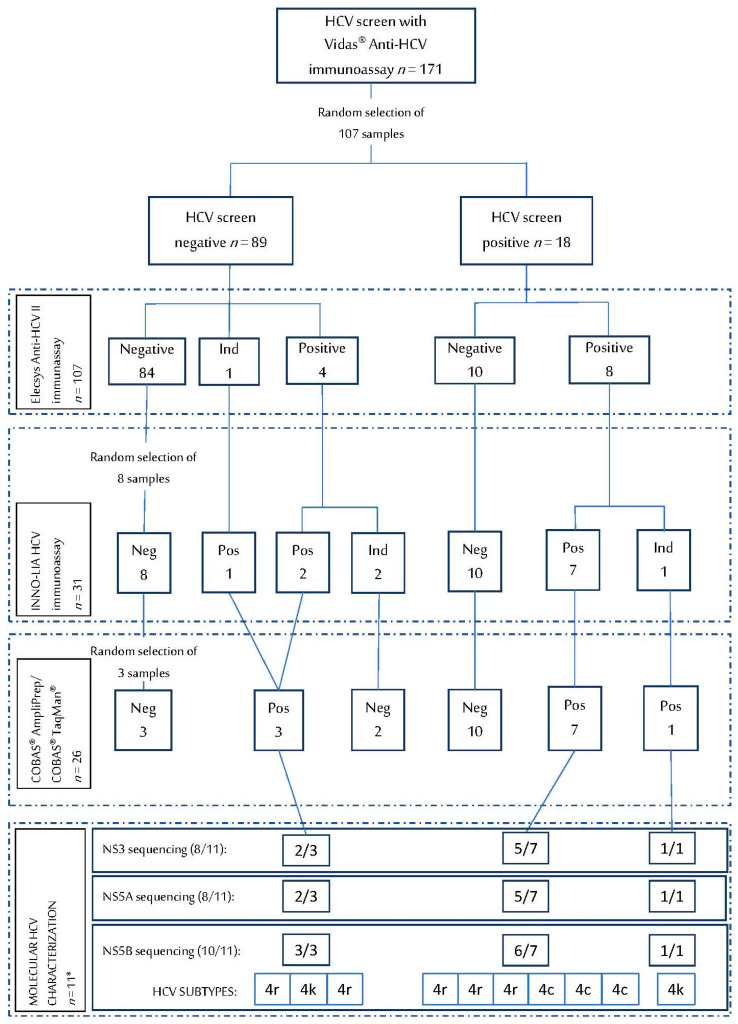
Summary of results obtained with serological and molecular techniques using dried blood spots (DBS) collected in Kinshasa (DRC). * Note: Resistance testing was attempted on all samples from viraemic patients (*n* = 11). Three samples with HCV viral load below 1000 IU/mL could not be amplified for NS3 and NS5A, but two of them could be subtyped by NS5B sequencing.

**Table 1 diagnostics-11-00522-t001:** Primer pairs for HCV GT4 NS5A, NS5B, and NS3 regions.

PRIMER	SEQUENCE	Position *
NS5A-4ckr_F	GVCTCCAYAAGTGGATCAAYGA	6146–6167
NS5A-4ckr_R	GAACCTGRCAGGGRCACTTGAT	6598–6619
VHC-NS5A2F	TGGGSTTYKCVTATGAYACCCG	8174–8195
VHC-NS5A2R	GARTACCTRGTCATAGCCTCCG	8549–8570
VHC-NS5B2F	GAYACCCGCTGYTTYGACTC	8188–8207
VHC-NS5B3R	CATAGCCTCCGTGAAGRCTC	8540–8559
NS3-4ckr_F1	AGGYTRGGCAATGARATAYTGCT	3283–3305
NS3-4ckr_R1	GARGGGTTRAGCACBAGCAC	4021–4040
NS3-4ckr_F2	GGAGRCTYCTTGCYCCCAT	3339–3357
NS3-4ckr_R2	GGRACYTTGGTGCTYTTGCC	3973–3992

HCV-NS5A2F (outer forward), HCV-NS5A2R (outer reverse), HCV-NS5B2F (inner forward), HCV-NS5B3R (inner reverse), NS5A-4ckr_F (forward), NS5A-4ckr_R (reverse), NS3-4ckr_F1 (outer forward), NS3-4ckr_R1 (outer reverse), NS3-4ckr_F2 (inner forward), NS3-4ckr_R2 (inner reverse). * Reference strain: FJ462439.

**Table 2 diagnostics-11-00522-t002:** Characteristics of three serological assays for hepatitis C diagnosis on DBS. Sensitivity, specificity, positive predictive value (PPV), and negative predictive value (NPV) (95% confidence interval) for VIDAS^®^ Anti-HCV (bioMerieux), Elecsys^®^ Anti-HCV II (Roche) and INNO-LIA HCV (Fujirebio).

	Vidas^®^ Anti-HCV (*n* = 107)	Elecsys Anti-HCV II (*n* = 107)	INNO-LIA HCV (*n* = 31)
Sensitivity (*n*/N)	8/13	13/13	13/13
% (95% CI)	61.5 (31.6–86.1)	100 (75.3–100)	100 (75.3–100)
Specificity (*n*/N)	84/94	94/94	18/18
% (95% CI)	89.4 (81.3–94.8)	100 (96.1–100)	100 (81.5–100)
Positive Predictive Value (*n*/N)	8/18	13/13	13/13
% (95% CI)	44.4 (21.5–69.2)	100 (75.3–100)	100 (75.3–100)
Negative Predictive Value (*n*/N)	84/89	94/94	18/18
% (95% CI)	94.4 (87.4–98.1)	100 (96.1–100)	100 (81.5–100)

**Table 3 diagnostics-11-00522-t003:** Baseline amino acids found at the main resistance-associated substitution (RAS) positions in NS3 region.

Subtype	Sample ID	NS3 RAS Codons					
		41	56	80	155	156	168
4r	CUN-8	Q	Y	Q	R	A	D
4r	CUN-112	Q	Y	Q	R	A	D
4r	CUN-319	Q	Y	Q	R	A	D
4r	CUN-369	Q	Y	Q	R	A	D
4c	CUN-26	Q	Y	Q	R	A	D
4c	CUN-66	Q	Y	Q	R	A	D
4c	CUN-285	Q	Y	Q	R	A	D
4k	CUN-71	Q	Y	Q	R	A	D

**Table 4 diagnostics-11-00522-t004:** Baseline amino acids found at the main RAS positions in NS5A region.

Subtype	Sample ID	NS5A RAS Codons				
		28	29	30	31	32
4r	CUN-8	**V**	P	R	**L**	P
4r	CUN-112	**I**	P	R	**L**	P
4r	CUN-319	**I**	P	R	**L**	P
4r	CUN-369	**F**	P	**S**	**L**	P
4c	CUN-26	L	P	R	M	P
4c	CUN-66	L	P	R	M	P
4c	CUN-285	L	P	R	M	P
4k	CUN-71	L	P	R	**L**	P

Note: Substitutions associated with reduced susceptibility to NS5A inhibitors are highlighted in bold.

**Table 5 diagnostics-11-00522-t005:** Genetic polymorphisms in NS3, NS5A, and NS5B regions among hepatitis C virus (HCV) strains circulating in Kinshasa (DRC).

Subtype	Sample ID	NS3 Mutations	NS5A Mutations	NS5B Mutations
4r	CUN-8	R26K, V134T, A181S	V8I, E10D, **I28V**, V56I, L108F	None
4r	CUN-112	R26K, M147L, V151A, A181S	L108F	C316H, V321I
4r	CUN-319	R26K, R130K, I132L, S133A, A181S	T17A, **P58S**, L108F, Y127F	H267Y, Y276H, C316H, V321I, E327G
4r	CUN-369	R26K, T95A, V107I, H110Q, A181S	E10D, T17A, I28F, R30S, V56T, N62S, K68R, L108V, R123Q	K270R, E327A
4r	CUN-157	N/A	N/A	None
4c	CUN-26	**T122N**	H6W, R44K, T56R, I74M, T75A, T83M, V121I, I130V	A252V, R309K
4c	CUN-66	L14F, **T122N**	H6W, R44K, T56R, T75A, T83M, V121I, I130V	A252V, R309K
4c	CUN-285	L14F, K92S, F105Y, H110R, M147K	H6R, R44K, T75V, E117D, V121I, I130V	H267Y, K333Q
4k	CUN-71	G15S, I71V, E95A, R98T, I107V, L127I, S181F	D10E, T17S, R44K, T83I, V101I, D117E, V121I, S127Y	S312T, T324A
4k	CUN-202	N/A	N/A	A252V, L262K, Y276L, V300T, T303A, K309R, C316N, T324A, E327D, R335A

Note: Polymorphisms at sites associated with resistance are represented in bold. Mutations found in two or more patients are underlined. N/A: Not available. Reference strains: FJ462436 (4c), EU392173 (4k), FJ462439 (4r).

## Data Availability

The data presented in this study are openly available in Harvard Dataverse at https://doi.org/10.7910/DVN/RUJTPA (accessed on 14 March 2021).

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
