# Peer review of "HCV Diagnosis and Sequencing Using Dried Blood Spots from Patients in Kinshasa (DRC): A Tool to Achieve WHO 2030 Targets"

_diagnostics, 2021, doi:10.3390/diagnostics11030522_

Round 1

Reviewer 1 Report

Thanks for considering and addressing the points raised in my original review, I hope they were reasonable and aided the study! I am happy with the synopsis of these changes and the updated manuscript with no additional comments to add. Well done for conducting and writing this work up!

Author Response

Dear reviewer, thank you very much for your previous comments to improve the manuscript and I hope that the article can be useful for those researcher working on HCV in Africa and worldwide.
Yours sincerely,
Gabriel Reina

Reviewer 2 Report

The presented paper is aimed to analyze diagnostic assays for HCV based on dried blood spots specimens collected in Kinshasa, Congo, and to characterize genetic diversity of the virus within a group of mainly HIV positive patients. Material and Methods are clearly described and sufficiently explained. Results and Discussion are very up to date. HCV subtype 4r has been the most prevalent among the patients. The authors concluded that the WHO eradication plan will not be achieved by 2030 if people living in low and middle-income countries cannot access diagnostic tests and subsequent treatment. There will be an overall benefit to publishing this work. The material is very interesting, original and significant. The English language is appropriate and understandable.

Author Response

Dear reviewer, thank you very much for your previous comments to improve the manuscript and I hope that the article can be useful for those researcher working on HCV in Africa and worldwide.
Yours sincerely,
Gabriel Reina

This manuscript is a resubmission of an earlier submission. The following is a list of the peer review reports and author responses from that submission.

Round 1

Reviewer 1 Report

The manuscript presented tackles a very important issue – how to diagnose and appropriately treat HCV infection in sub-Saharan Africa (SSA) or LMICs in general, where in-country laboratory infrastructure is lacking, in line with WHO targets. Dried blood spots are a valuable analyte in such circumstances and although this is well explored in the literature, surprisingly little has been recorded with regards to performance from SSA countries and extending analysis to sequencing. The majority of the samples were assessed only in terms of antibody reactivity, but again given the evidence here and elsewhere, it is important to evaluate existing assays against the genotypes more prevalent in SSA. Pan-genotypic NS5b primers were able to detect the majority of samples determined RNA positive by a commercial assay, and this information was used to iterate further NS3 and NS5A primers to assess antiviral susceptibility. The interpretation supports the conclusions and helps to build the literature base for HCV in SSA.

The report is generally well written but could do with some minor improvements to English throughout – although nothing currently affects the comprehension of the manuscript.

A more general point to make that while it is laudable that investigations into SSA HCV are being undertaken, it was a little disappointing to see only one individual on the author list affiliated with a DRC institution despite all the samples being collected there. Should more Congolese medical staff be authors on this paper, rather than just in the acknowledgements? This would help strengthen Western links with SSA institutes, which could help in a small way towards the efforts of HCV elimination.

Specific points

Abstract / methods / elsewhere – please define and restate the cohort population better. These pts are likely to be a high-risk population due to very high HIV co-infection rates, so when the various statistics are quoted, it needs to be clear that these are unlikely to be representative of the population as a whole. Otherwise citing literature may misattribute seroprevalence etc.

Also note whether Urban / rural, Kinshasa only etc

Title: spots from patients in Kinshasa (DRC). A tool to 3 achieve WHO 2030 targets

                Colon to separate statements, rather than full stop?

Line 20 ‘HCV antibodies detection’  antibody. An instance where English grammar could be improved (amongst others, some noted below, probably more too).

Line 80 ‘there are often no suitable infrastructures to perform them’

Line 203 ‘(11.2%). Ten samples tested with the INNO-LIA-HCV Score assay, were positive (9.3%). ‘

line 220 ‘in Figure 1; HCV viral load values ranged between 369 and 423645 IU/mL.’

                does this account for dilution factors? 2 x 70ul blood spots extracted and eluted in what volume? 5ul of this was used, thus only a fraction of a ml used. Add to methods 2.3 & 2.4.2?

Table 6.

                Note relative to what ref strain (from Geno2pheno)

‘CUN-157               NA           NA           - ‘

                Unify format i.e. use NA (N/A?) or –, not both?

261 Several studies have focused on the reliability of DBS as HCV markers.

                Rewrite – markers doesn’t seem correct to me? Maybe ‘DBS as an analyte for HCV diagnosis’?

289 ‘the sensitivity of the assays evaluated did no exhibit significant’     not

296 A high rate of active HCV infection was detected among Congolese patients (77%), similar to 74- 88% reported by Layden et al in SSA [23].

                An example of where to explicitly state cohort / pt group, as noted above – otherwise suggests random population screen.

308 ‘The NS5B region was chosen to characterize the genotype and subtype of our patients as there were

310 ‘were determined for 90.9% of sequenced samples and 4c/4k/4r subtypes were highly prevalent.’

                Just prevalent, as that was the full extent of the genotyping

314  ‘as there were obtained 8/11 viable sequences’ rewrite

342 ‘A RAS pattern, previosuly described for ledipasvir’

                Previously

346 ‘28, 30 and 31 [27], however its availability in SSA may be far.’ rewrite

348 ‘In this study, there were no any amino acid alteration at this position associated with reduced susceptibility.’ rewrite

376 ‘The recommendation in UK point out the investigation’ rewrite

390 population. Second, no next-generation sequencing

                Define: (NGS)

393 ‘Despite these limitations, DBS sample has demonstrated’

                Sampling?

396 ‘In conclusion, WHO eradication plan by 2030 will’

                In conclusion, the WHO eradication by 2030 plan will

403 prevalent strain among HCV patients in Kinshasa.

                Perhaps add population size to emphasise importance ‘…Kinshasa, a city of X million people’.

General discussion point – DAA efficacy / contraindication in HIV-coinfected individuals an issue worth mentioning?

Reviewer 2 Report

The background:

- I would suggest a few changes to content and language, including updating data from 2015, accounting all treatment failures to RASs, the notion that genotypic resistance is useful for personalised treatment,. I'd suggest a greater focus and detail on SSA context, including access to pan-genotypic treatment etc, rather than global numbers.

The Aims / novelty:

  • I think the Aims might be better placed on the feasibility of conducting testing from DBS, rather than a thorough Sens / spec analysis, because (i) a performance evaluation of sens / specificity is not possible without gold standard conducted from paired plasma sample, (ii) assays were not modified / validated for DBS sample type
  • Aim 2 (genotype distribution in DRC) cannot be achieved with these numbers and sampel selection (n = 11?)
  • The novelty lies in the demonstration that DBS samples can be used to evaluate HCV sequences as with partient management, or even as a surveillance tool to understand the genotype distribution. The numbers are too small to conclude these results represent the DRC broadly.

Methods:

  • The authors need to include a description of the study design to be sure if it’s a prospective / retrospective study.
  • Patient / sample inclusion criteria are not clear, although the primary study is referred to by name. The authors should include more detail so it is clear if these patients are treatment naïve / DAA naïve / treatment failures / screening / post-treatment etc.
  • Were any paired plasma samples available within the analysis to allow a direct comparison of methods compared to a reference test? It is not clear that a reference test is used to allow a sensitivity / specificity study
  • The authors might refer to this guideline, including helpful checklists on how to structure the paper and some requirements for inclusion.
  • https://www.equator-network.org/reporting-guidelines/stard/
  • If paired samples were not collected as part of this study, can the authors access the Reference lab result based on plasma samples, and group according the reference test conducted (eg. if the reference test from one lab was Roche) as performance will need to be compared to one test.
  • If the clinical results from the central lab is not available as a Reference test, have any of the methods used in the paper been sufficiently validated that could justify their use as a “gold standard” reference test?
  • serological tests, where DBS-adapted protocols followed for analysis (eg. haematocrit calculations / differences in volume loaded for each assay) or the routine method by manufacturer for plasma followed)? This detail needs to be added.
  • The authors should provide a more information on if the methods used been validated by other labs that can be referred to in the literature as published methods, or are they in-house?
  • The numbers of samples tested and numbers for positive / indeterminate, should be reported as a result and inform the flow chart. The numbers tested as described here were a little confusing
  • The authors will need to refer to data or published methods to support LOD of "270 IU/mL of DBS in EDTA plasma 147 and 210 IU/mL in serum. RNA is quantifiable over the value of 860 IU/ml of eluted DBS". Was the 860 IU/mL used as a threshold in the analysis? This detail needs to be added.
  • Resistance testing - selection criteria for this testing needs to be described (all viraemic patients? Above a xx IU/mL? The numbers tried / failed and reasons why need to go into the flow chart in the results section.
  • Resistance testing - should we suggest they also use an on-line tool, I'm not sure if this paper is a widely accepted method considering information on resistance is always being updated. On-line tools may provide a more up to date analysis. If this is not possible, the authors may compare the method in the paper used and see if there are any major differences in the on-line tools and describe these.
  • The authors should consider including a description of the analysis plan, including the comments above to ensure this is clear before presenting the results.
  • Some language could be improved throughout.

Results:

  • Flow chart needs to be redrawn as it is currently difficult to follow. The flow chart needs to include a description of the number of samples collected, and number of samples test by each method, the numbers of results generated for each method and the numbers included in analysis. This will allow the readers ot understand exactly what results are used to evaluate each index case and the numbers on a reference method (plasma / gold standard if this is the case) to conduct a sensitivity and specificity analysis. The reasons for the drop in numbers at each step should be described (eg. VL too low as mentioned in sequence results section, insufficient samples, samples too old? invalid results etc). examples in the literature and
  • The analysis for the serology results Table 3, as it stands, does not seem to be a traditional diagnostic sensitivity / specificity analysis of each index test relative to a reference tests.
  • Likewise, without any information on how the ses / spec was conducted for the RNA results, the readers are not able to interpret the meaning of the results.
  • Consider if the storage conditions, and different durations may have impacted results? Perhaps include this as a limitation.
  • Needs a clear description of the patient characteristics, separate from sampel characteristics.
  • - RNA detection - how is the RNA sens / specificity calculated? What Gold standard is this against?
  • Resistance results
    • how were the bases called, categorised as a mutation, using what method as a cut-off?
    • There seems to be a reference to baseline mutations, where their treatment failure samples in this selection? Are their longitudinal samples to compare with? This needs more clarity.
    • Table 6 mutations associated with resistance (underlined) - what information is this based on, eg. clinical trial results, or in-vitro data? Please provide clear references to justify thet inclusion of mutations considered significant

Discussion - this was not read, but I’m very happy to review and provide comments on the next version once the methods / results are clearer.

Please know all of these comments aim to strengthen the paper and provide an opportunity for the work to be published. I do hope the authors find it helpful.